# Health status of free-ranging ring-necked pheasant chicks (*Phasianus colchicus*) in North-Western Germany

J. Liebing[1], I. Völker[2], N. Curland[1], P. Wohlsein[2], W. Baumgärtner[2], S. Braune[3], M. Runge[3], A. Moss[4], S. Rautenschlein[5], A. Jung[5], M. Ryll[5], K. Raue[6], C. Strube[6], J. Schulz[7], U. Heffels-Redmann[8], L. Fischer[8], F. Gethöffer[1], U. Voigt🄳[1], M. Lierz[8], U. Siebert🄳[1]*

1 Institute for Terrestrial and Aquatic Wildlife Research, University of Veterinary Medicine Hannover, Hannover, Germany, 2 Department of Pathology, University of Veterinary Medicine Hannover, Hannover, Germany, 3 Lower Saxony State Office for Consumer Protection and Food Safety (LAVES), Food and Veterinary Institute Braunschweig/Hannover, Hannover, Germany, 4 Lower Saxony State Office for Consumer Protection and Food Safety (LAVES), Food and Veterinary Institute Oldenburg, Oldenburg, Germany, 5 Clinic for Poultry, University of Veterinary Medicine Hannover, Hannover, Germany, 6 Institute for Parasitology, Centre for Infection Medicine, University of Veterinary Medicine Hannover, Hannover, Germany, 7 Institute for Animal Hygiene, Animal Welfare and Farm Animal Behaviour, University of Veterinary Medicine Hannover, Hannover, Germany, 8 Clinic for Birds, Reptiles, Amphibians and Fish, Justus Liebig University Giessen, Giessen, Germany

* ursula.siebert@tiho-hannover.de

**Data Availability Statement:** Data are available in Supplement S1 data.pdf

**Funding:** "Funding was partly provided by Lower Saxony Ministry of Food, Agriculture and

## Abstract

Being a typical ground-breeding bird of the agricultural landscape in Germany, the pheasant has experienced a strong and persistent population decline with a hitherto unexplained cause. Contributing factors to the ongoing negative trend, such as the effects of pesticides, diseases, predation, increase in traffic and reduced fallow periods, are currently being controversially discussed. In the present study, 62 free-ranging pheasant chicks were caught within a two-year period in three federal states of Germany; Lower Saxony, North Rhine-Westphalia and Schleswig-Holstein. The pheasant chicks were divided into three age groups to detect differences in their development and physical constitution. In addition, pathomorphological, parasitological, virological, bacteriological and toxicological investigations were performed. The younger chicks were emaciated, while the older chicks were of moderate to good nutritional status. However, the latter age group was limited to a maximum of three chicks per hen, while the youngest age class comprised up to ten chicks. The majority of chicks suffered from dermatitis of the periocular and caudal region of the head (57–94%) of unknown origin. In addition, intestinal enteritis (100%), pneumonia (26%), hepatitis (24%), perineuritis (6%), tracheitis (24%), muscle degeneration (1%) and myositis (1%) were found. In 78% of the cases, various Mycoplasma spp. were isolated. *Mycoplasma gallisepticum* (MG) was not detected using an MG-specific PCR. Parasitic infections included Philopteridae (55%), *Coccidia* (48%), *Heterakis/Ascaridia* spp. (8%) and *Syngamus trachea* (13%). A total of 8% of the chicks were Avian metapneumovirus (AMPV) positive using RT-PCR, 16% positive for infectious bronchitis virus (IBV) using RT-PCR, and 2% positive for haemorrhagic enteritis virus (HEV) using PCR. All samples tested for avian encephalomyelitis virus (AEV), infectious bursal disease virus (IBDV) or infectious laryngotracheitis virus

Consumer Protection, the State Agency for Nature, Environment and Consumer Protection of North Rhine-Westphalia and the Ministry of Energy, Agriculture, the Environment and Rural Areas of Schleswig-Holstein. No additional external funding was received for this study."

**Competing interests:** The authors have declared that no competing interests exist.

(ILTV) were negative. The pool samples of the ten chicks were negative for all acid, alkaline-free and derivative substances, while two out of three samples tested were positive for the herbicide glyphosate. Pheasant chick deaths may often have been triggered by poor nutritional status, probably in association with inflammatory changes in various tissues and organs as well as bacterial and parasitic pathogens. Theses impacts may have played a major role in the decline in pheasant populations.

## Introduction

The original distribution area of the ring-necked pheasant (*Phasianus colchicus*) ranged from the Black Sea over the dry areas of Central Asia to the East of Asia to South Korea and Siberia [1]. The Romans introduced the pheasant to Europe around 500 AD, from where it spread through regular release throughout Central and Western Europe [2]. According to current published data, the pheasant mainly prefers structurally semi-open land, using trees and hedges as cover, and also occupies adjacent sparse forests and reedy areas [3]. Most pheasants seek shelter under trees to be protected from natural predators. However, some subspecies spend the night on the ground or among dense reeds. Their resting places during the day are usually well-hidden hedges, where sand-baths are taken in carved hollows [1].

Adult pheasants mainly feed on plants, consuming different parts of the plant such as seeds, berries, tubers, root shoots and leaves, as well as green sprouts. However, on occasions, their diet is supplemented by animal protein, preferably in the form of insects [1]. For chicks, smaller ground-level insects are especially important during the first weeks of life. They feed on a variety of species of insects such as spur cicadas (Delphacidae), bugs (Heteroptera), sawfly wasps (Tenthredinidae) and butterfly caterpillars (Lepidoptera larvae) [1, 4, 5]. This diversity is particularly important for a healthy growth [6, 7]. For example, a diet based only on aphids can lead to delayed plumage development due to inadequate amino acid supply [8].

In Germany, the pheasant is a typical soil-breeding bird of the agricultural landscape. The main part of the German population is found in southwest Lower Saxony, North Rhine-Westphalia and Schleswig-Holstein. The population level reached its plateau between 1960 and 1970 in Lower Saxony. During this period, the hunting bag statistics (State registered numbers of hunting animals, in this case pheasants), i.e. the absolute number of pheasants killed, amounted to approximately 300,000 pheasants in Germany [9]. In the severe winter of 1970 and the following wet spring of 1971, the population of pheasants and many other wild living animals declined [9, 10]. The hunting bag was reduced to an average of about 80,000 pheasants and declined further. Not only was the pheasant population subjected to this decline, but also that of many other farmland birds [11, 12, 13]. Around 2007/2008, the population showed another severe decline of unknown cause. In Germany, the Renewable Energy Sources Act (Erneuerbare-Energie-Gesetz: Renewable Energy Sources Act describes the implementation of ecological energy generation in Germany) amendment of 2004 with an advancement in biogas, triggered the doubling of corn cultivation. Consequently, huge areas of fallow land disappeared in Lower Saxony [14]. The contributory factors to the ongoing decline in the pheasant population, such as the effects of pesticides, infectious agents, predation, increasing traffic and human populations as well as reduced fallow periods, are currently the subject of controversial discussion among different stakeholders [15, 16, 17, 18]. Some authors see a correlation between the changes in agriculture and the decline in the populations of many farmland birds [19, 20]. In the third week of life, 70% of the chicks' diet consists of insects. Gradually, the

insect percentage is reduced. From the sixth week of life, the diet is similar to that of adult birds. Previous studies [21, 22] associated the decline in the number of many farmland birds with the use of insecticides. If chicks are unable to find a sufficient number of insects during the first weeks of life, they have to search a larger range of their habitat, which can lead to malnutrition and weakening. Thus, harmless ubiquitous pathogens may have negative effects on chicks [23, 24, 25, 26, 27, 28].

Investigations carried out led to the assumption that there is no specific epidemic infectious agent currently circulating in the adult pheasant population [29]. Many hunters report that especially the number of chicks has declined, with more older birds making up the hunting bag. However, the authors found serological evidence of certain viruses (infectious bronchitis virus (IBV), avian encephalomyelitis virus (AEV) and infectious bursal disease virus (IBDV)) which typically cause chick mortality. These pathogens infected adult and young pheasants, but the pathogenicity in chick and subadult populations is considerably more serious than in adult birds [29, 30]. In addition, other factors may weaken the population and pathogens become more important. Based on these findings, our study focused on pheasant chicks up to eleven weeks of age. Previous studies on pheasants indicated that the most sensitive age class for infectious diseases was pheasant chicks, possibly due to a higher susceptibility [30, 31]. The aim of our research was to assess the health state of free-living pheasant chicks in order to check the animals for lesions indicative of infections or toxic substances. The findings should contribute to understanding the causes for the decline in the pheasant population in North-Western Germany.

## Materials and methods

### Animals

In 2014 and 2015, the Institute for Terrestrial and Aquatic Wildlife Research (ITAW), University of Veterinary Medicine Hannover, Foundation, Hannover and the Wildlife Research Institute, State Office for Nature, Environment and Consumer Protection of North Rhine-Westphalia caught free-living Ring-necked Pheasant chicks from Lower Saxony (Cuxhaven, Grafschaft Bentheim, Emsland, Osnabrück, Vechta), North Rhine-Westphalia (Coesfeld, Warendorf) and Schleswig-Holstein (Dithmarschen) to assess the health state by means of pathological, microbiological, virological, parasitological and toxicological investigations. The caught chicks were grouped into three age classes (ac) based on the feather markings of the hand-wings. Age class one (ac1) included chicks up to three weeks of age, ac2, chicks from four to six weeks of age and ac3, chicks older than six weeks and up to 11 weeks.

### Catching chicks

An animal experiment permit was obtained from the responsible veterinary office of the Lower Saxony State Office for Consumer Protection and Food Safety (LAVES) (permit number: 33.14-42502-04-14/1486). The study areas comprised 11 hunting regions with 15 traps in Lower Saxony (Hemmoor, Meppen, Neuenkirchen, Osten, Strücklingen, Vechta, Wilsum,)11 regions with 10 traps in North Rhine-Westphalia (Ahlen, Dülmen, Lippstadt, Welte) and 4 districts with 4 traps in Schleswig-Holstein (Warwerort) (Fig 1). The catching period lasted from May until August. In 2014, the investigated chicks were three to 11 weeks old. In 2015, the age of the chicks varied from one-day-old to eleven-week-old chicks. At the age of 11 weeks, the young pheasants were considered as sexually mature. After the catch, the mother hen was released and at maximum, half of the chicks in the trap were taken for analysis (mostly one-three chicks at random). In 2014, the traps had a size of 2.3 $m^2$ and were covered with iron bars with a mesh-size of 1 $cm^2$. In 2015, the traps were slightly adapted based on experience

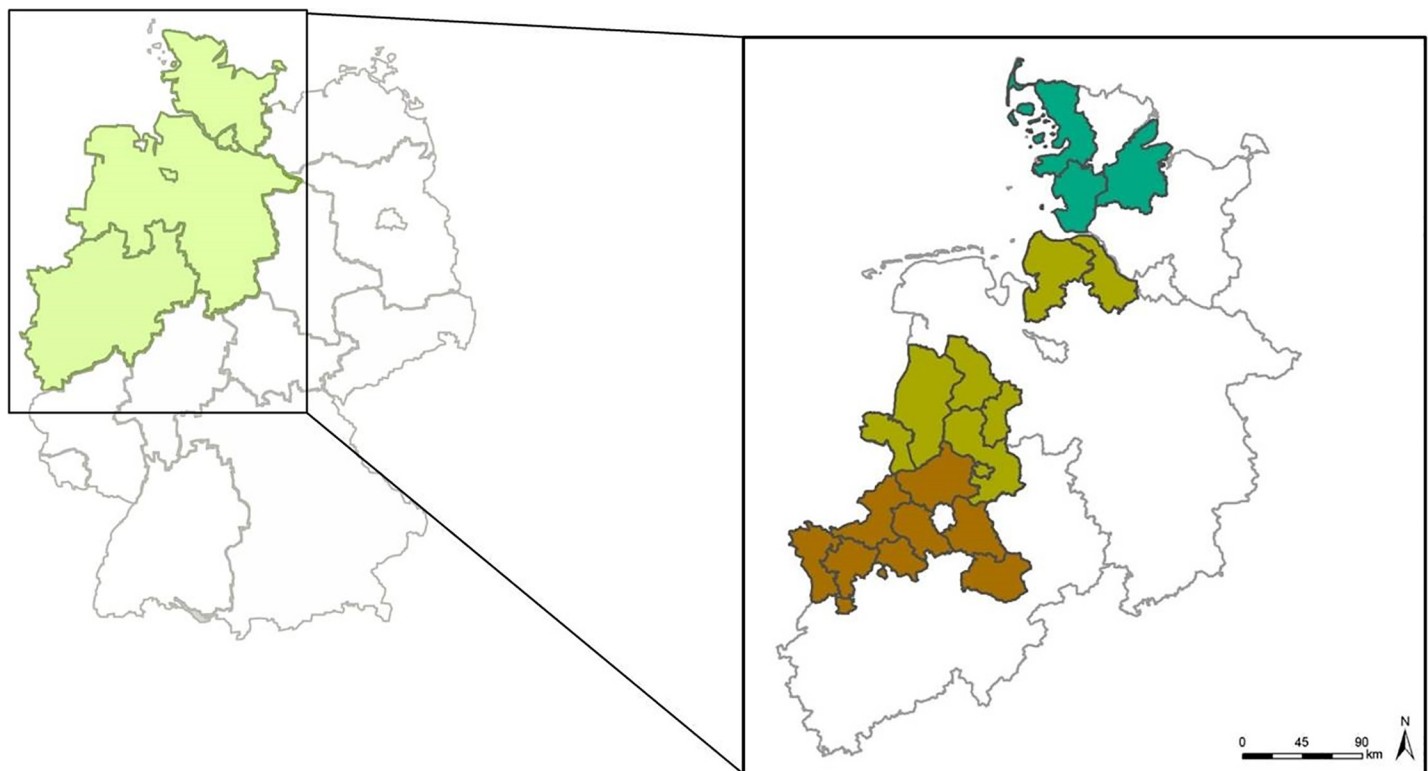

**Fig 1. Map of Germany with the study areas Lower Saxony, North Rhine-Westphalia and Schleswig Holstein, © GeoBasis-DE / http://www.bkg.bund.de, 2019.**

from 2014, using a cover made of loose polyethylene netting with a mesh-size of 1 cm$^2$. A piece of string was used as a trigger so that both trap doors closed when the chicks moved forward. Corn was used to attract the hen and her chicks. Afterwards, the chicks were transported alive to the University of Veterinary Medicine Hannover for examination. The time span from catch to examination took on average about five hours. In 2014, the chicks were stunned by a head blow and killed by exsanguination. In 2015, the chicks were euthanised with an intravenous injection of pentobarbital-sodium (Boehringer-Ingelheim, AG & Co. KG, Ingelheim, Germany).

## Nutritional condition score

The nutritional condition score was evaluated macroscopically by the thickness of the pectoral muscles and the body fat percentages as good, moderate, poor or cachectic. As described by Curland et al. [29], animals in a good body condition revealed a vast amount of fatty tissue within the thoracic and abdominal regions, whereas animals with a moderate body condition demonstrated reduced amounts of body fat tissue. Animals in a poor body condition possessed only low amounts of fat reserves, these frequently associated with pectoral muscle atrophy. In contrast, cachectic animals lacked fat reserves and exhibited serous atrophy of the coronal myocardial fatty tissue.

## Pathomorphology and histopathology

The necropsy was carried out in accordance with the standard protocol [31]. Representative samples of the following tissues and organs were collected, fixed in 10% neutral-buffered

formalin and routinely embedded in paraffin wax: the skin of the head and abdomen, skeletal muscle (*Musculus pectoralis*, *Musculus quadriceps*), ischiadic nerve, brachial plexus, nose with infraorbital sinus, eye with lacrimal gland, bone with bone marrow, trachea, thymus, thyroidal gland, lung, heart, liver, pancreas, spleen, kidney, crop, proventriculus, gizzard, intestine, adrenal gland, gonads, bursa of Fabricius and brain. Paraffin sections of 3–5 μm were stained with haematoxylin and eosin (HE) for histological examination. In selected cases, periodic acid Schiff (PAS) reaction, Ziehl-Neelsen stain, Brown-Brenn stain and Turnbull's blue stain were performed [32].

## Parasitology

For parasitological examinations, samples of the small intestine were collected from all 62 chicks at necropsy. At the same time, skin and plumage of the chicks were macroscopically examined for ectoparasites. For coproscopical examination, the combined sedimentation-flotation method was performed: the faecal sample was filled into a tea strainer (mesh size 1 mm) and rinsed in a beaker with a jet of water. The filtrate containing helminth eggs and protozoan oocysts was allowed to sediment for 30 min. Afterwards, the supernatant was decanted and the sediment transferred to a 15-mL centrifuge tube filled with saturated zinc sulphate solution (ZnSO$_4$, specific gravity 1.30) and centrifuged at 450 x g for 5 min. The liquid surface was transferred onto a slide with a wire eyelet and examined microscopically. If at least one egg or oocyst was detected, the sample was classified as positive. A semiquantitative classification was applied using the following key: one-two eggs or oocysts were categorised as mild, six-ten eggs or oocysts as moderate, 11–20 eggs or oocysts as severe; if more than 20 eggs or oocysts were detected, the shedding intensity was classified as by mass [29].

## Virology and serology

The fresh samples, consisting of brain, trachea and caecal tonsils, as well as the bursa of Fabricius were placed in RNAlater® (Sigma-Aldrich Chemie GmbH, München, Germany). Samples were analysed by RT-PCR for avian metapneumovirus (AMPV), infectious bronchitis virus (IBV), avian encephalomyelitis virus (AEV) and by PCR for infectious bursal disease (IBDV) and infectious laryngotracheitis as described in [33, 34]. Serum was taken from all birds to check for antibodies against avian influenza virus (AIV) subtypes H5, H7 and H9. Eight additional liver samples from chicks with hepatitis were analysed for the presence of haemorrhagic enteritis virus (HEV) by PCR [29].

## Microbiological investigations for Mycoplasma

For microbiological investigations for *Mycoplasma*, 13 tracheal swabs, six tracheal tissue samples and three periorbital skin tissue samples were taken [29]. The samples were directly transferred to *Mycoplasma* cultivation medium (SP4).

**Detection of Mycoplasma by PCR.** For DNA extraction, swabs were soaked and rubbed in 350 μL phosphate buffered saline (PBS). Using the DNeasy ® Blood & Tissue Kit (Qiagen GmbH, Hilden, Germany) in accordance with the manufacturer's instructions, 100 μL of the liquid was taken for DNA extraction. For DNA extraction of tissue samples and the single colony subcultures, the fluid medium from culturing (2 mL) was centrifuged at 4000 x g for 45 minutes. The remaining pellet was incubated with 180 μL lysis buffer (ATL Buffer, Qiagen, GmbH) and 20 μL proteinase K (Qiagen GmbH) for two hours at 56˚C.

All samples and single colony subcultures were screened via *Mycoplasma*-genus-specific PCR (target: 16S rRNA gene sequence) for DNA of *Mycoplasma* spp. as described by [35] and modified [36]. From all single colony subcultures, an additional PCR (target: 16S-23S rRNA

sequence (Intergenetic Transcribed Spacer Region)) was performed [37]. Furthermore, all samples were examined via *Mycoplasma gallisepticum*-specific PCR [38]. The PCR products were sequenced by a commercial DNA sequencing service (LGC Genomics GmbH, Berlin, Germany). The sequences of the PCR products were aligned with the 16S rRNA gene and 16S-23S rRNA ISR sequences of *Mycoplasma* spp. in the NCBI database using BLAST (NCBI, Bethesda, MD, USA) algorithm [39].

**Mycoplasma culture.** The samples were cultured using SP4 liquid and agar media produced in house as described previously [34]. Each sample was immersed in the SP4 broth and afterwards removed and stored for further investigations. The broth was diluted (ten-fold dilution up to $10^{-2}$) and an aliquot of 50 μL each was transferred onto agar media. Both, liquid and solid media were incubated at 37°C with 5% $CO_2$ in a humidified environment for up to ten days. Broth was examined for colour change and agar plates for colony growth daily. In case of colour change, or after five days, an additional "subculture" on agar media was performed. In case of *Mycoplasma* growth, several single colony subcultures were performed at least twice in order to ensure pure species cultures. Each third single colony subculture was stored at -80° C until further investigation by molecular biological methods [36, 40].

## Toxicology

Liver samples of nine pheasants were screened for herbicide glyphosate and other pollutants (for details see S1 Table). Of these nine samples, one sample was taken from a ten-chick ratchet in ac1, while the remaining eight samples were single-samples from ac3 with liver or kidney inflammation. The samples were stored directly after autopsy at -80°C.

Toxicological samples (n = 10), 7 g liver pool samples, were used to detect substances by means of the gas chromatography-mass spectrometry (GC-MS) method (performed at the Institute of Pharmacology, Toxicology and Pharmacy of the Ludwig-Maximilians-University, Munich, Germany). The samples were tested for substances greater than 70 Dalton (D) (including pesticides, heavy metals, inorganic substances, mycotoxins and plant poison). Analyses of glyphosate, aminomethylphosphonic acid (AMPA), as well as a screening of more than 650 other pesticides by GC-MS and/or LC-MS/MS were performed by the DIN EN ISO/IEC 17025:2005-accredited laboratory Eurofins Sofia GmbH, Berlin, Germany. For analyses of glyphosate and AMPA in liver and other organs, the samples were homogenised, acidified and extracted. An aliquot of the extract was neutralised and derivatised by 9-fluorenyl-methoxycarbonylchloride. Measurement was performed by liquid chromatography-mass spectrometry (LC-MS/MS), quantification was performed by adding a known amount of standard sample directly to an aliquot of analysed sample. For pesticide screening in the liver and other organs, homogenised samples were extracted following the standard German method (§ 64 LFGB L 00.00–34 2010–09 (modified) (in short: fat extraction, clean-up using gel permeation chromatography, measurement by GC-MS, quantification by adding a known amount of standard material of representative matrix) or § 64 LFGB L 13.04–5 2013–08 (in short: extraction using acetonitrile and water, measurement by LC-MS/MS, quantification by adding a known amount of standard material of representative matrix) (Eurofins GfA Lab Service GmbH, Hamburg, Germany).

## Results

During the two-year-period, a total of 62 chicks were caught: 29 birds in Lower Saxony, 27 in North Rhine-Westfalia and six in Schleswig-Holstein. Fourteen chicks were allocated to ac1, 16 to ac2 and 32 to ac3. Of the investigated animals, 34 were female, 17 male and for 11 chicks, macroscopic gender estimation was unknown; histological samples were not taken.

**Table 1. Nutritional status of the investigated pheasants (number of pheasants).**

| Nutritional status | ac1 | ac2 | ac3 |
|---|---|---|---|
| Good | 0 | 9 | 25 |
| Moderate | 0 | 7 | 6 |
| Poor | 12 | 0 | 1 |
| Cachectic | 2 | 0 | 0 |

## Nutritional status

The nutritional status in ac1 was predominantly poor (n = 12; 85.7%) or cachectic (n = 2, 14.3%). In ac2, approximately nine (56.3%) of the chicks were well fed and seven (43.8%) were moderately fed. The majority of birds in ac3 were well fed (n = 25, 78.1%), some were moderately fed (n = 6, 18.8%) and one chick (3.1%) was in a poor body condition (Table 1).

## Pathomorphological findings

To an excessive amount, mild to severe cutaneous abrasions with feather loss, lacerations and/or subcutaneous haemorrhages of the head were noticed in one out of two chicks (50%) in ac1, nine out of 16 chicks (56%) in ac2 and 11 out of 20 chicks (55%) in ac3 trapped with the TT1. One out of 12 animals (8%) in ac1 and nine out of 12 individuals (75%) in ac3 that had been trapped with TT2 were more mildly affected by those lesions. Histological examination of the skin from the head revealed various types of inflammatory alterations which occurred solely or concurrently in one individual (Table 2). In all age classes and independent of trap type, mainly perivascular predominantly lympho-histiocytic dermatitis admixed with occasional heterophils and plasma cells of varying degrees was present (Fig 2). This type of inflammation was in some cases accompanied by follicular aggregations of lymphocytes sometimes with secondary follicle formation (ac3, TT2). In addition, ulcerative (Fig 3), occasionally necrotising, suppurative and pustular inflammatory changes were found more often in chicks trapped with TT1 than with TT2, these in many cases being associated with dermal and/or subcutaneous haemorrhages of varying degrees. Only a few animals showed no cutaneous alteration in this localisation. The abdominal skin of the chicks was rarely affected by perivascular dermatitis; single individuals were mainly affected by lymphohistiocytic or pustular dermatitis.

Crops, glandular stomachs and gizzards were variably filled (Table 3). However, it should be noted that the chicks had spent up to five hours in the traps. The mentioned parts of the digestive tract contained grains, green food, and, inside the gizzard, grit stones. In one chick (7%) in ac1, in seven chicks (44%) in ac2 and in ten chicks (31%) in ac3, the quality of the intestinal content was associated with hyperaemic intestinal mucosa and perianal attachment of faeces suggestive of catarrhal enteritis.

**Table 2. Morphological findings in the skin of the head of pheasants (number of pheasants).**

| | ac1 | | ac2 | | ac3 | |
|---|---|---|---|---|---|---|
| | TT1 | TT2 | TT1 | TT2 | TT1 | TT2 |
| Total number of pheasant chicks | 2 | 12 | 16 | 0 | 20 | 12 |
| Perivascular, mainly lympho-histiocytic dermatitis | 2 | 7 | 10 | 0 | 19 | 8 |
| Ulcerative, occasionally necrotising dermatitis | 0 | 0 | 6 | 0 | 1 | 2 |
| Suppurative dermatitis | 0 | 1 | 6 | 0 | 4 | 0 |
| Pustular dermatitis | 0 | 3 | 7 | 0 | 5 | 1 |
| Granulomatous dermatitis | 0 | 0 | 1 | 0 | 0 | 0 |
| Without inflammatory changes | 0 | 3 | 1 | 0 | 1 | 2 |

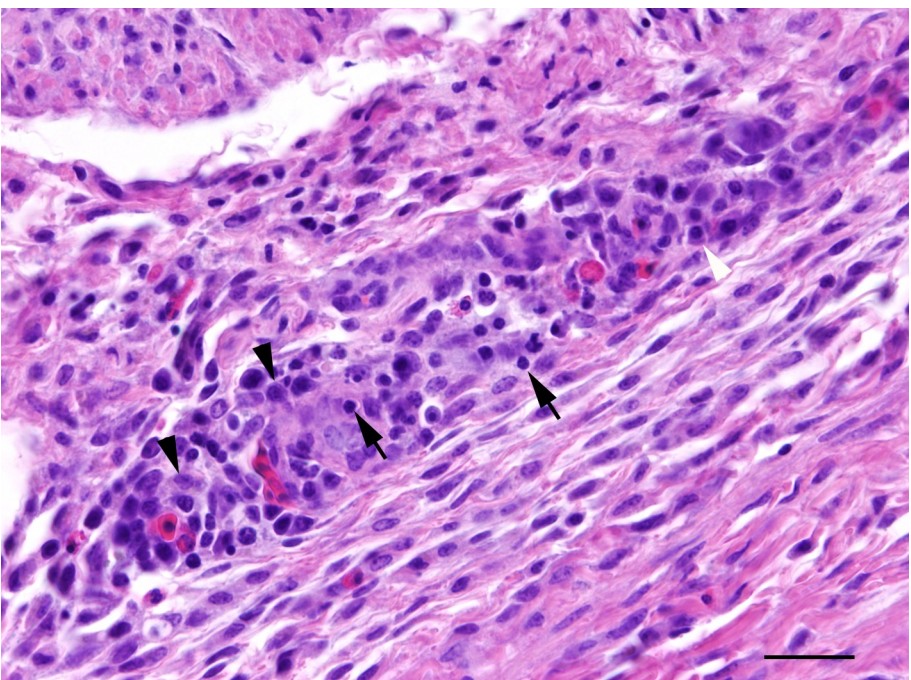

**Fig 2. Pheasant chick with moderate perivascular dermatitis of the skin of the scalp with infiltration of lymphocytes (arrows), macrophages (black arrowheads) and few plasmacells (white arrowhead); HE, bar = 60 μm.**

Histologically, one animal (3%) in ac3 showed focally severe ulcerative stomatitis at the gums and one chick in ac3 focally moderate lympho-histiocytic ingluvitis. In single chicks in ac3, nematodes without reactive inflammatory changes were found in the crop. Furthermore, single individuals showed erosive and heterophilic inflammation of the gizzard, occasionally associated with intralesional nematodes which were not differentiated here. The intestinal mucosa in all animals showed a mild to moderate infiltration with eosinophils, lymphocytes and a few plasma cells. In eight out of 16 chicks (50%) in ac2 and in four out of 32 chicks (13%) in ac3, reproduction stages of protozoal organisms, most likely *Coccidia* sp., were found histologically within the epithelium (Fig 4). Predominantly mild focal lymphohistiocytic hepatitis was observed in one out of 14 chicks (7%) in ac1, in four out of 16 chicks (25%) in ac2, and in ten out of 32 chicks (31%) in ac3. Single individuals in ac3 showed multifocal granulomatous hepatitis with severe acute coagulation necrosis (Fig 5). In eight out of 14 chicks (57%) in ac1, a mild to severe diffuse fatty change in hepatocytes was present.

Nematodes with the morphology consistent with *Syngamus trachea* were found in the trachea of none of the 14 chicks in ac 1, in five out of 16 chicks (31%) in ac2 and in ten out of 32 (31%) chicks in ac3. Histologically, tracheal parasitism in most animals was associated with multifocal lympho-histiocytic, occasionally granulomatous or ulcerative tracheitis of variable extent. In single animals, subepithelial lymphoid follicles were found. In the lung, focal or multifocal interstitial, mild to moderate lymphohistiocytic pneumonia was observed in one out of 14 chicks (7%) in ac1, in three out of 16 chicks (19%) in ac2, and in six out of 32 chicks (19%) in ac3. Focal or multifocal, mild to moderate granulomatous, occasionally necrotising pneumonia was present in three individuals (19%) in ac2 and in two individuals (6%) in ac3. One animal in ac3 suffered from severe suppurative to necrotising pneumonia. There was no evidence of viral, bacterial, fungal or parasitic agents in these lungs, even in the histological special stains. Hyperplasia of bronchus-associated lymphoid tissue was noticed in two chicks (13%) in

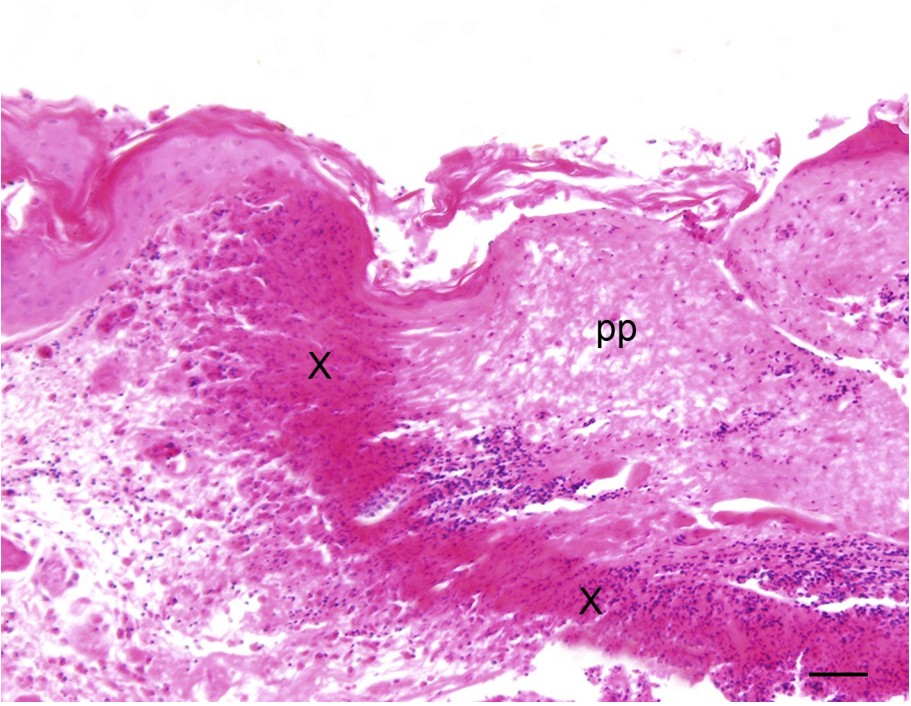

**Fig 3. Pheasant chick with severe ulcerative dermatitis of the skin of the head with accumulation of cellular debris (X) and proteinaceous exudate (pp); HE, bar = 120 μm.**

ac2 and in three chicks (9%) in ac3. Numerous lungs showed acute haemorrhages. The kidneys displayed focally mild interstitial infiltrations consisting mainly of lymphocytes and macrophages in two chicks (13%) in ac2 and five chicks (16%) in ac3. Independent of the age class, a mostly moderate diffuse infiltration with plasma cells was observed in almost all examined lacrimal glands. Miscellaneous findings included focally mild lymphocytic myocarditis in one chick (3%) in ac3 (Fig 6), focally moderate lymphohistiocytic perineuritis (*N. ischiadicus*) in one chick in both ac2 (19%) and ac3 (3%), severe subacute hyaline degeneration of skeletal muscles with histiocytic infiltration in one chick (3%) in ac3, focal chronic suppurative

**Table 3. Amount of ingested food in the gastric tract finding during necropsy.**

| Amount of ingested food | ac1 | ac2 | ac3 |
|---|---|---|---|
| **Crop** | | | |
| Moderate | 0 | 0 | 0 |
| Marginal | 0 | 0 | 0 |
| No content | 14 | 16 | 32 |
| **Glandular stomach** | | | |
| Moderate | 0 | 0 | 0 |
| Marginal | 0 | 0 | 4 |
| No content | 14 | 16 | 25 |
| **Gizzard** | | | |
| Moderate | 1 | 1 | 16 |
| Marginal | 2 | 10 | 10 |
| No content | 11 | 5 | 6 |

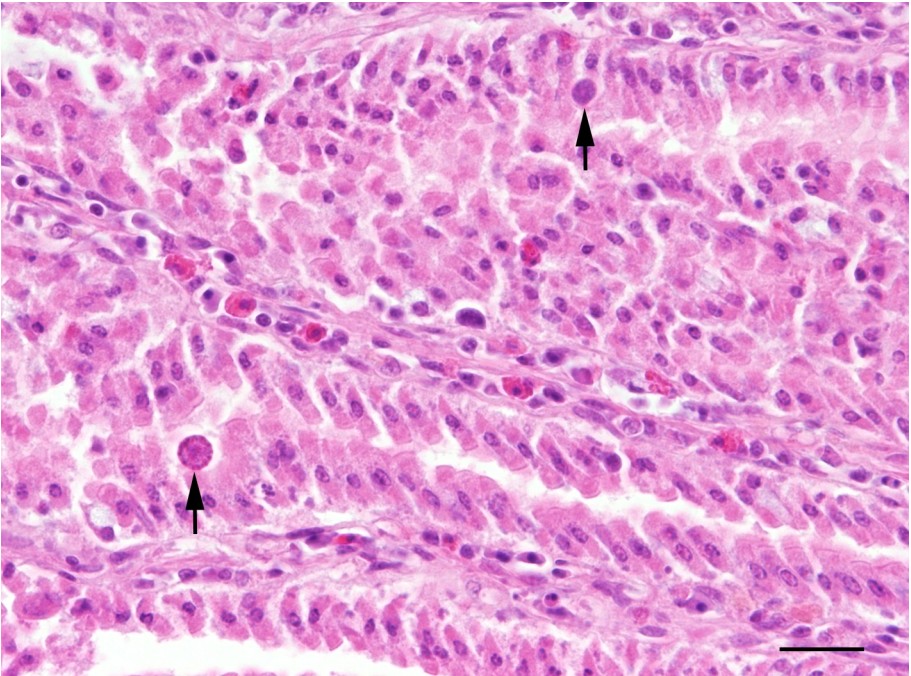

**Fig 4. Pheasant chick with mild intestinal coccidiosis characterised by single protozoal microorganisms (arrows) in enterocytes; HE, bar = 60 μm.**

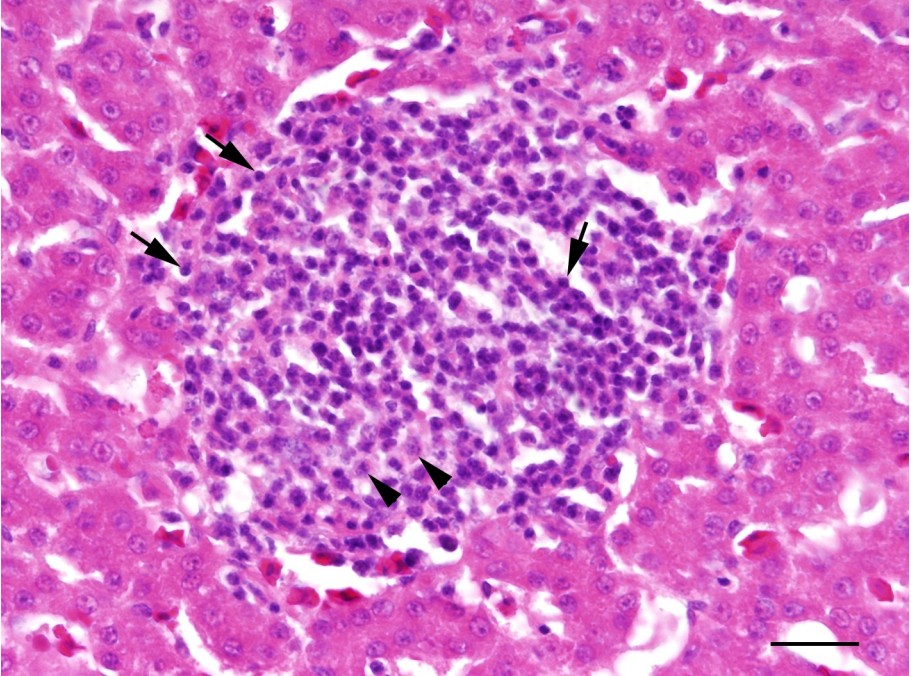

**Fig 5. Pheasant chick with focal mild hepatitis characterised by infiltration of lymphocytes (arrows) and macrophages (arrowheads); HE, bar = 60 μm.**

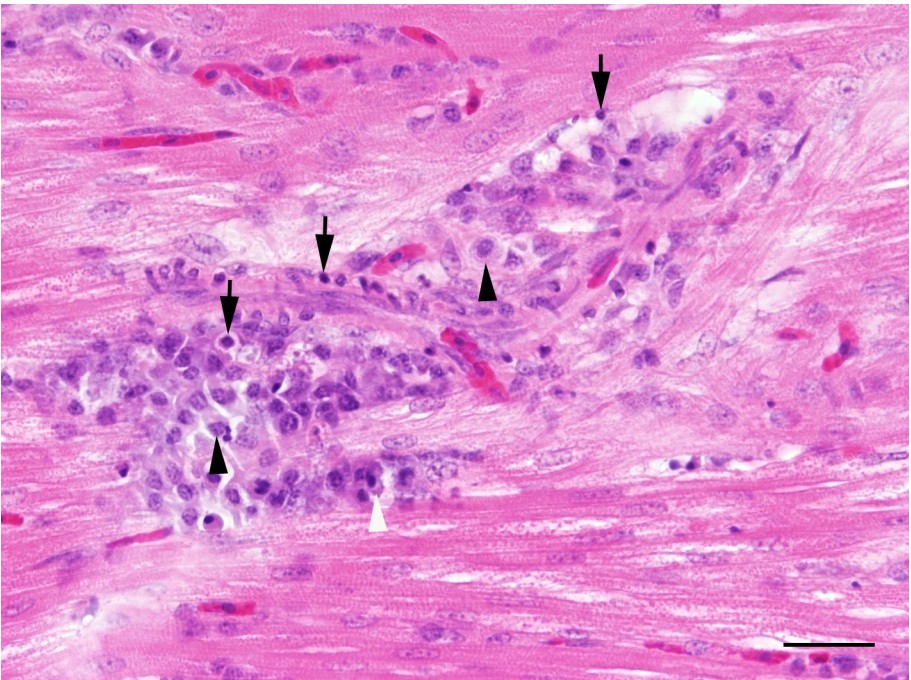

**Fig 6. Pheasant chick with focal mild myocarditis characterised by infiltration of lymphocytes (arrows), macrophages (black arrowheads) and plasmacells (white arrowhead); HE, bar = 60 µm.**

myositis in another chick (3%) in ac3, and single protozoal cysts, most likely sarcosporidia sp., in the skeletal musculature of two individuals (6%) in ac3 without inflammatory changes. In many brains, perivascular and parenchymatous haemorrhages were observed without reactive changes.

## Virology

Of all 62 chicks tested, 8% of the chicks were positive for avian metapneumovirus (AMPV) using RT-PCR, 16% positive for infectious bronchitis virus (IBV) using RT-PCR, and 2% for haemorrhagic enteritis virus (HEV) using PCR. None of the 37 chicks tested for HEV were positive using PCR. Tracheae of 33 chicks and caecal tonsils of ten birds were tested by the Coronavirus-RT-PCR and were negative for the respective virus. All samples tested for avian encephalomyelitis virus (AEV), infectious bursal disease virus (IBDV), or infectious laryngo-tracheitis virus (ILTV) were negative.

## Bacteriology

All examined samples tested *negative for Mycoplasma gallisepticum (MG) via MG-specific-PCR. In 19 out of 21 samples*, *Mycoplasmal* DNA was detected using *Mycoplasma*-genus-specific PCR (target: 16S rRNA gene sequence). In 15 out of 19 samples, culturing was success-fully performed and various *Mycoplasma* spp. were isolated and identified using PCR, target-ing the 16S-23S rRNA sequence (Intergenetic Transcribed Spacer Region). The following *Mycoplasma* spp. were frequently isolated: *M. gallinaceum*, *M. glycophilum*, *M. iners* and *M. pullorum*.

## Parasitology

During examination of skin and plumage, ectoparasites of the family Philopteridae (n = 34) were detected. Coproscopical examination for endoparasites revealed *Coccidia* (n = 30), *Heterakis/ Ascaridia* spp. (n = 5) and *Syngamus trachea* (n = 8), while five chicks were negative for endoparasite stages. In eight birds, coinfections with two different parasites (*Coccidia* and *Syngamus trachea*) were found.

## Toxicology

The pool samples of the ten chicks were completely negative for all acid, alkaline-free and derivative substances as listed in S1 Table which summarises the substances tested and the detection limits. Two out of three samples tested for the herbicide glyphosate were positive (0.044 mg/kg, 0.095 mg/kg).

## Discussion

Since the 1970s, a population decrease in ring-necked pheasants has been observed. Especially in 2007/2008, the population decline intensified [41]. The present investigation revealed that the randomly trapped pheasant chicks displayed inflammatory lesions in different organs. In association with environmental stressors and a depleted nutritional status, these health changes may increase pheasant chick mortality, thus contributing to the population decrease.

The dermatitis detected was often of a non-purulent character, mostly perivascularly accentuated with different cellular compositions of gradual variable infiltrations by lymphocytes, plasma cells and macrophages. Especially on the head, this alteration additionally displayed pustular and lymphocytic inflammation. It was possibly itch- or parasite-induced. Avian pox was excluded due to lack of pathognomonic and histological changes [42]. These alterations occurred in chicks as young as one or two days of age. Six out of 12 chicks (50%) in ac1 already showed these alterations, with different types and degrees of inflammation. *M. gallisepticum* (MG) is an important etiological differential diagnosis, especially inducing periocular dermal swelling with lymphocytic inflammation [43]. However, this pathogen was ruled out by the investigations. Nevertheless, various *Mycoplasma* spp. were isolated in 15 out of 21 (71.4%) of the investigated chicks. However, the role of these *Mycoplasma* spp. as a potential cause of periorbital skin alterations in pheasants is still unclear, but should be considered in following investigations in pheasants. As some birds were RT-PCR positive for IBV or AMPV, it has to be elucidated further whether these viruses may have contributed to these lesions, as they are known to be respiratory disease associated. The inflammations might be itch induced following insect or tick bites. Furthermore, the head injuries with lacerations and haemorrhages resulted from catching caused by the chicks jumping against the iron bars of the traps. These injuries did not appear anymore after exchanging these bars for loose nylon mesh.

A total of 65% of the 26 pneumonia cases were of an eosinophilic character and were most likely caused by *Syngamus trachea* in ten cases. All cases of granulomatous inflammation were free of acid-fast bacteria as shown by the Ziehl-Neelsen stain. Therefore, the cause of this granulomatous inflammation remains unknown. Other possible agents able to induce pneumonia, bronchopneumonia, tracheitis and bronchitis were not detected.

A prevalent eosinophilia tracheitis (94%) occurred in almost all cases in connection with detected parasites at different stages including *Coccidia*, *Heterakis/Ascaridia* spp. and *Syngamus trachea*. The degrees of inflammation were mainly mild up to moderate so that the clinical relevance is rather subordinate.

The proventriculitis can have many origins. A histologically similar disease, that of gizzard erosion in broilers is often caused by an interaction between vitamin deficiency, fungal

infections and stress situations after consuming mycotoxins. With periodic acid Schiff reaction (PAS) and Brown-Brenn stain, fungi, Gram-positive and Gram-negative bacteria were excluded [44]. As AMPV, IBDV, coronavirus and siadenovirus were excluded by PCR, a viral cause is relatively unlikely. Marek's disease is doubtful as well due to the lack of other typical organ changes [45]. It is possible that the birds may have been exposed to mycotoxins or pesticides that caused proventriculitis. Based on localisation, size and shape of the eggs found in the proventriculus, a nematode-infection with *Dyspharynx nasuta* probably resulted [46].

The inflammation of the livers showed lymphocytic and lymphohistiocytic characters. The causes for these inflammatory changes are manifold and may include infectious as well as non-infectious agents. In three cases, the inflammation was granulomatous and necrotising. Using Ziehl-Neelsen stain, acid-resistant bacteria were excluded. Differential diagnoses for granulomatous and necrotising hepatitis include toxic, ischemic or infectious causes [47, 48]. In the presented investigations, only a limited number of samples could be investigated for pesticides. Therefore, it is difficult to directly link pathological findings to any of the investigated chemicals. Further investigations are needed to elucidate the role played by pesticides in the declining pheasant populations as their habitat is regularly exposed to different chemicals used in agriculture.

## Conclusions

The main findings in the study were the poor nutritional status in the younger age groups and the increasing occurrence of various inflammation when the birds were ageing. As no direct cause for the inflammation was found and the inflammation affected various organs, it might be more a sign of various pathogens affecting the chicks. This seems to be more a sign of a weakened immune system, unable to defeat facultative pathogenic organisms. This is in line with the poor nutrition status, which triggers the development of diseases. No suspected virus infection was detected though. Virus infections cannot be ruled out completely as a cause as viruses obviously circulate in the adult pheasant population and infected chicks die quickly. Therefore, such cases were not among the sampled animals as the study focused on live chicks which still followed the hen. Due to predation, decomposition and vegetation in the field, diseased pheasants are difficult to retrieve for health examinations and therefore were not included here. Concerning parasites, low *Coccidian* infections can be regarded as desirable to build up a protective immunity against reinfections. However, intestinal changes of the chicks show that *Coccidia* sometimes considerably damage the intestinal mucosa due to severe infections, which may lead to a reduction in nutrient uptake. Furthermore, a severe *Syngamus sp*. infection can occlude the tracheal lumen, resulting in suffocation of the chicks, or their general condition deteriorates to such an extent that they become easy prey for predators. All these findings point to an effective complexity that either chicks die of starvation or their immune system becomes weakened. It seems that when the effects of maternal antibodies slowly diminish and the chicks have to mobilise their own immune system, the chicks become weakened as their immune system is not sufficiently developed. Also, it is known from poultry that the development of the immune system is influenced by nourishment and that malnutrition can negatively influence the immune system [31, 49, 50]. This hypothesis has not been confirmed yet in pheasants. Not only pheasants, but also many other farmland birds have to cope with the intensive agricultural landscape. This change in habitat and the use of pesticides make it increasingly difficult to find insects that are vital for the chicks during their first weeks of life. A whole concatenation of circumstances could be explained by this connection; namely, the rather poor nutritional status, a possibly weakened immune system and the increased susceptibility to diseases. Additionally, it is possible that the chicks are easier prey for predators due to

various inflammations or poor physical condition, too. Also, the weather can have a greater influence.

## Supporting information

**S1 Table. Toxicological investigation of substances and limits of detection.** Highlighted in bold are the substances found in the pheasant samples.
(DOCX)

**S1 Data.**
(PDF)

## Acknowledgments

We wish to thank the hunting associations of Lower Saxony, North Rhine–Westphalia and Schleswig-Holstein for supporting the study. Furthermore, special thanks go to the laboratory personnel for their excellent technical assistance in the laboratory investigations.

## Author Contributions

**Conceptualization:** N. Curland, F. Gethöffer, U. Voigt, M. Lierz, U. Siebert.

**Data curation:** J. Liebing.

**Formal analysis:** J. Liebing, I. Völker, P. Wohlsein, S. Braune, M. Runge, A. Moss, S. Rautenschlein, M. Ryll, J. Schulz, U. Heffels-Redmann, L. Fischer.

**Funding acquisition:** C. Strube, U. Siebert.

**Investigation:** J. Liebing, I. Völker, P. Wohlsein, S. Braune, M. Runge, A. Moss, S. Rautenschlein, A. Jung, M. Ryll, K. Raue, J. Schulz, U. Heffels-Redmann, L. Fischer.

**Methodology:** J. Liebing, I. Völker, P. Wohlsein, S. Braune, M. Runge, A. Moss, S. Rautenschlein, A. Jung, M. Ryll, K. Raue, C. Strube, J. Schulz, L. Fischer.

**Project administration:** U. Siebert.

**Resources:** J. Liebing, P. Wohlsein, W. Baumgärtner, M. Runge, S. Rautenschlein, C. Strube, J. Schulz, M. Lierz, U. Siebert.

**Software:** J. Liebing.

**Supervision:** U. Siebert.

**Validation:** J. Liebing, P. Wohlsein, S. Rautenschlein, C. Strube, M. Lierz, U. Siebert.

**Visualization:** J. Liebing, P. Wohlsein.

**Writing – original draft:** J. Liebing, P. Wohlsein, S. Rautenschlein, A. Jung, K. Raue, C. Strube, J. Schulz, U. Heffels-Redmann, L. Fischer, M. Lierz, U. Siebert.

**Writing – review & editing:** J. Liebing, I. Völker, N. Curland, P. Wohlsein, W. Baumgärtner, S. Braune, M. Runge, A. Moss, S. Rautenschlein, A. Jung, M. Ryll, K. Raue, C. Strube, J. Schulz, U. Heffels-Redmann, L. Fischer, F. Gethöffer, U. Voigt, M. Lierz, U. Siebert.

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
