## [Decision Letter · Decision Letter 0]

10 Feb 2020

PONE-D-19-33260

Health status of free-ranging ring-necked pheasant chicks (Phasianus colchicus) in North-Western Germany

PLOS ONE

Dear Dr Siebert

Thank you for submitting your manuscript to PLOS ONE. After careful consideration, we feel that it has merit but does not fully meet PLOS ONE’s publication criteria as it currently stands. Therefore, we invite you to submit a revised version of the manuscript that addresses the points raised during the review process.

Many thanks for submitting your manuscript to PLOS One

Apologies for the delay in returning comments to you. I invited over 30 people to review this and struggled to get reviewers due to Christmas etc.

However, I managed to get three reviewers who are experts in the field to review the manuscript.

The reviewers have made some good comments to help you to improve the manuscript

If you could write a detailed response to reviewers that would be most helpful as I plan to return the manuscript to the same reviewers

I wish you every success with your revisions

Many thanks

Simon

We would appreciate receiving your revised manuscript by Mar 26 2020 11:59PM. To enhance the reproducibility of your results, we recommend that if applicable you deposit your laboratory protocols in protocols.io, where a protocol can be assigned its own identifier (DOI) such that it can be cited independently in the future. For instructions see: http://journals.plos.org/plosone/s/submission-guidelines#loc-laboratory-protocols

We look forward to receiving your revised manuscript.

Kind regards,

Simon Russell Clegg, PhD

Academic Editor

PLOS ONE

1. In your Methods section, please provide additional location information of the trapping sites, including geographic coordinates for the data set if available.

4. Thank you for including the following funding information within your acknowledgements section of your manuscript; " We would also like to thank the Lower Saxony Ministry of Food, Agriculture and Consumer Protection, the State Agency for Nature, Environment and Consumer Protection of North Rhine-Westphalia and the Ministry of Energy, Agriculture, the Environment and Rural Areas of Schleswig-Holstein for financially supporting the study. "

Reviewers' comments:

Reviewer's Responses to Questions

**Comments to the Author**

1. Is the manuscript technically sound, and do the data support the conclusions?

Reviewer #1: Yes

Reviewer #2: Partly

Reviewer #3: Partly

2. Has the statistical analysis been performed appropriately and rigorously? 

Reviewer #1: N/A

Reviewer #2: N/A

Reviewer #3: I Don't Know

3. Have the authors made all data underlying the findings in their manuscript fully available?

Reviewer #1: Yes

Reviewer #2: Yes

Reviewer #3: Yes

4. Is the manuscript presented in an intelligible fashion and written in standard English?

Reviewer #1: Yes

Reviewer #2: No

Reviewer #3: Yes

5. Review Comments to the Author

Reviewer #1: This is a very well written interesting manuscript describing the health statues of free-ranging phesants. Neverhteless, as it is presented this article is better suited for another type of journal such as Wildlife Diseases Journal (WDJ). The audinece in of WDJ would be very intersted in the authors findings.

Reviewer #2: • Inconsistent writing

• Grammatical issues (sentences not necessarily making sense or flowing nicely, not very easy to read in some cases)

• Flow from one point to another is lacking (jumps from one point to next)

• Few writing errors (no gaps between full stops, random letters throughout paragraphs)

• Sometimes no gaps between paragraphs (doesn’t look professional) and isn’t consistent with the rest of the document

• Inconsistent with giving data and results (sixteen in 54, 12 in 67 – should stick to one format etc.)

• Inconsistent explanations for doing something or none at all

Questions:

1. What acid and alkaline-free derivative substances?

2. What other factors may weaken the population?

3. Why focus on birds up to 11 weeks of age?

4. Exactly how many chicks were taken? (at maximum half isn’t an amount unless you first specify the amount caught)

5. Why did the animals have to die for you to test this?

6. Nutritional condition score previously described where?

Comments:

Don’t agree with methodology – seems counterintuitive and backwards to what they opened the paper with, and the decline was the entire reason behind the study.

Reviewer #3: This is a very interesting, novel article, offering some fantastic insights into the health of free ranging wild birds which is always tricky to undertake. I think that the authors have done a generally good job, however I do have some concerns about the manuscript which are detailed below.

Could I request that you put line numbers in please, as it makes reviewing much easier?

Abstract

Various Mycoplasma spp (Mycoplasma needs to be in italics)

Same line, space between isolated. Mycoplasma

Line starting Heterakis- Starting the line after with a percentage is not good English- consider rewording the sentence

Also avian doesn’t need a capital

Introduction

I think your opening line could do with some more details. You mention original distribution- maybe say where it was? When? Where its native to? Where it has spread etc?

3rd line- comma after release

Natural predator line and some species- maybe consider merging

Line 2- second paragraph- comma after leaves

Paragraph 3- Explain what hunting bag statistics are

Not only was the pheasant subject to decline….- consider rewording as it doesn’t make sense

In Germany, the Renewable energy act- reword this sentence as it is unclear.

At the sixth week of life ….- reword this sentence as it doesn’t make sense

Paragraph 4- line 4- typically caused by

Line below- this could merge with the report of many hunters- reword this as it doesn’t make sense

Line below that, remove the I from the start of the sentence

These pathogens infected – consider rewording this sentence as it is unclear

Materials and Methods

You mention the feather markings of the hind wings- can you insert a bit of the importance of this?

You mention the animal testing permit- does that cover all the regions which you tested? Catching chicks, line 6, comma after 2015

You say catch varied from 1 day old to 11 week old chicks- did this age of capture affect the results in any way?

You then mention the mother hen, which is the first time that she was released. I find the catching section a little hard to follow and in a slightly strange order

You also say that half of the chicks were chosen- how was this done? Randomly?

Remove full stop after Hannover

Line below- put 5 in words

The section on nutritional condition score may benefit from a bit of expansion so its clear as this is a rarely published area?

Pathomorphology section- first line is not a full sentence - it is because of the refenrec eso maybe reword it?

The first time the muscles are mentioned they probably want to be in full, rather than abbreviated to M …

Insert and between Fabricius and brain

I think its often written as 450 x g

You talk about parasite egg counts- were the infecting parasites identified?

Virology- coma after tonsils in line 1

Avian metapneumovirus doesn’t need capitals

PCR line doesn’t make sense as it is- it is because of the refenrec eso maybe reword it?

Is there any reason why you didn’t test for avian influenza?

Microbiology- how were the swabs chosen? As the different tissues will likely lead to different results

Mycoplasma culture- SP4 liquid and media need a manufacturer- maybe also some detail on the type of agar media?

Liquid and solid media were inoculated- which ones?

In the case of colour change, or after five days, an additional …. (add in commas)

Mycoplasma needs capitals and italics

Mycoplasma PCR- PBS needs a manufacturer

Not good to start a sentence with 100ul

Again Mycoplasma needs capitals and italics

First line of second paragraph doesn’t make sense- it is because of the refenrec eso maybe reword it?

Results

Why were 11 chicks gender unknown? Too young to tell?

You mention some haemorrhages in the pathomorphological section- could these have been caused during trapping?

2nd paragraph of pathomorphological section- line 5. Comma after ac3

Line 2 of paragraph 3- comma after ac3

You mention nematodes seen in the pheasants- were these further identified?

Line on intestinal mucosa displayed in all animals- reword as unclear

Coccidia needs capitalising and italicising

Were the protozoans not further identified either?

Sarcocporidia- needs capitalising and italicising

Virology section- comma after tested in first line

Line 2- remove positive by

Bacteriology section – you did 13 tracheal swabs from the 23 chicks- why was this number smaller and not one done from each?

Third line- remove that

You say via molecular biological methods- which ones- explain them please so we know

Comma after fifteen on last line of first paragraph

First line of 2nd paragraph- remove were

2nd line of 2nd paragraph- Mycoplasma wants capitals and italics

Parasitology- Line 3- spaces around the Ascaridia spp

Discussion

Line 1- observed has a typo

Line two- decline increased sounds a bit strange- consider rewording

Last two lines of paragraph one do not make sense and need rewording

Paragraph 2- line 7- comma after alterations

Line 7, remove as from start of sentence and put M gallisepticum in full

Line 9. Full stop after [42] and start a new sentence

Line 11- mycoplasma in italics

Penultimate line of paragraph 2- last but not least sounds colloquial., maybe finally?

Paragraph 3, not good to start it with a number

Paragraph 5- resulting in Dyspharynx nasuta – doesn’t make sense, consider rewording

Also last reference of paragraph 5 in capitals and incorrectly cited

And references in paragraph 6 are the same- maybe a glitch with your reference management software

Conclusions- line Line 7- hyphen seems inappropriate

End of first conclusion paragraph- maybe state that this hypothesis hasn’t been confirmed yet in pheasants

Table 4- I feel that this is a bit too big for a paper. Could this be supplementary?

It may also be nice to pull out any which were increased in the pheasants- maybe highlight them in bold?

6. PLOS authors have the option to publish the peer review history of their article (what does this mean?). If published, this will include your full peer review and any attached files.

Reviewer #1: No

Reviewer #3: No

---

## [Author Response · Author response to Decision Letter 0]

12 May 2020

Comments from the editors and reviewers:

-Reviewer 1

This is a very well written interesting manuscript describing the health statues of free-ranging pheasants. Nevertheless, as it is presented this article is better suited for another type of journal such as Wildlife Diseases Journal (WDJ). The audience of WDJ would be very interested in the authors' findings.

Thank you for the comment. We think the Plos One is a journal reaching out to a large community including researchers interested in Wildlife Diseases.

-Reviewer 2

• Inconsistent writing. 

 Has been improved on.

• Grammatical issues (sentences not necessarily making sense or flowing nicely, not very easy to read in some cases). 

 Has been improved on.

• Flow from one point to another is lacking (jumps from one point to next). Unfortunately, sometimes it could not be avoided. 

 Has been changed as far as possible.

• Few writing errors (no gaps between full stops, random letters throughout paragraphs). 

 Has been corrected.

• Sometimes no gaps between paragraphs (doesn’t look professional) and isn’t consistent with the rest of the document. 

 Has been corrected.

• Inconsistent with giving data and results (sixteen in 54, 12 in 67 – should stick to one format etc.). 

 Has been corrected.

• Inconsistent explanations for doing something or none at all

 The manuscript has been proofread by a native speaker before resubmission. 

1. What acid and alkaline-free derivative substances?

These substances are listed in Table 4. Screening by GC-MS and/or LC-MS/MS was performed by the DIN EN ISO/IEC 17025:2005-accredited laboratory Eurofins Sofia GmbH, Berlin, Germany.

2. What other factors may weaken the population?

The effects of pesticides, infectious agents, predation, increasing traffic, change of agriculture and human populations are the main reasons for the decline in the pheasant population after research and internal investigations. These factors are equally discussed for other farmland birds. 

3. Why focus on birds up to 11 weeks of age?

Thank you for the comment. Based on previous published data (Curland et al. 2018) of the Institute for Terrestrial and Aquatic Wildlife Research, University of Veterinary Medicine Hannover, Foundation, Germany, we found pathogens which are important in this age class. So, as a consequence, this study focuses on pheasant chicks up to eleven weeks of age. This has been modified. 

4. Some of Exactly how many chicks were taken? (at maximum half isn’t an amount unless you first specify the amount caught amount caught)

One to three chicks were taken. The hatches usually comprised no more than six chicks. The chicks were chosen at random. In one case, the hunter had caught the entire clutch without the hen. Since the chicks cannot survive without a hen, we took the entire hatches. 

5. Why did the animals have to die for you to test this?

The aim of our research was to assess the health state of free-living pheasant chicks and investigate the animals for inflammatory and any other lesions as well as infectious diseases to understand the high mortality rate in this age class. Pheasant chicks are too small to work with based on biopsies of different organs. 

6. Nutritional condition score previously described where?

In the study by Curland et. al 2018, the assessment of the nutritional status was described in detail. We have now added this information. 

Don’t agree with methodology – seems counterintuitive and backwards to what they opened the paper with, and the decline was the entire reason behind the study.

Has been adapted and hope it is now better explained. 

-Reviewer 3

This is a very interesting, novel article, offering some fantastic insights into the health of free ranging wild birds which is always tricky to undertake. I think that the authors have done a generally good job, however I do have some concerns about the manuscript which are detailed below.

1. Could I request that you put line numbers in please, as it makes reviewing much easier? Done

2. Abstract

Various Mycoplasma spp (Mycoplasma needs to be in italics). Done

3. Same line, space between isolated. Mycoplasma. Done

4. Line starting Heterakis- Starting the line after with a percentage is not good English- consider rewording the sentence. Done

Also avian doesn’t need a capital. Done

Introduction

5. I think your opening line could do with some more details. You mention original distribution- maybe say where it was? When? Where its native to? Where it has spread etc?. Has been changed

6. 3rd line- comma after release. Done

7. Natural predator line and some species- maybe consider merging

Done

8. Line 2- second paragraph- comma after leaves. Done

9. Paragraph 3- Explain what hunting bag statistics are. Done

10. Not only was the pheasant subject to decline….- consider rewording as it doesn’t make sense. Done

11. In Germany, the Renewable energy act- reword this sentence as it is unclear. Has been changed.

12. At the sixth week of life ….- reword this sentence as it doesn’t make sense. Done

13. Paragraph 4- line 4- typically caused by. Done

14. Line below- this could merge with the report of many hunters- reword this as it doesn’t make sense. Has been changed

15. Line below that, remove the I from the start of the sentence. Done

16. These pathogens infected – consider rewording this sentence as it is unclear. Has been changed

Materials and Methods

17. You mention the feather markings of the hind wings- can you insert a bit of the importance of this? Has been changed

18. You mention the animal testing permit- does that cover all the regions which you tested? Has been added

19. Catching chicks, line 6, comma after 2015. Done

20. You say catch varied from 1 day old to 11 weeks old chicks- did this age of capture affect the results in any way? Has been added

21. You then mention the mother hen, which is the first time that she was released. I find the catching section a little hard to follow and in a slightly strange order. Has been added

22. You also say that half of the chicks were chosen- how was this done? Randomly? Has been changed

23. Remove full stop after Hannover. Done

24. Line below- put 5 in words. Done

25. The section on nutritional condition score may benefit from a bit of expansion so its clear as this is a rarely published area? Has been added

26. Pathomorphology section- first line is not a full sentence - it is because of the refenrec eso maybe reword it? Has been changed

27. The first time the muscles are mentioned they probably want to be in full, rather than abbreviated to M …. Done

28. Insert and between Fabricius and brain. Done

29. I think its often written as 450 x g. Done

30. You talk about parasite egg counts- were the infecting parasites identified?

In the Material and Methods section, we talk about egg / oocyst counts in terms of semiquantitative classification of parasite egg or oocyst counts in different categories (mild, moderate, severe and by mass). Of course, we have identified the infecting parasites on the level allowed by egg or oocyst morphology. This is given in the results section, where we describe the identified parasites: coccidia as well as Ascaridia or Heterakis spp. (mentioned species cannot be reliably differentiated by egg morphology) and the lungworm Syngamus trachea.

31. Virology- coma after tonsils in line 1. Done

32. Avian metapneumovirus doesn’t need capitals. Done

33. PCR line doesn’t make sense as it is- it is because of the refenrec eso maybe reword it? Done

34. Is there any reason why you didn’t test for avian influenza? 

As described in lines 217-218, we tested for avian influenza (serum was taken from all birds to check for antibodies against avian influenza virus (AIV) subtypes H5, H7 and H9).

35. Microbiology- how were the swabs chosen? As the different tissues will likely lead to different results. 

At the beginning of the investigation, tissue samples from the trachea were taken and at a later stage it was changed to swabs for organisational reasons. Periorbital skin tissue was added to verify if lesions were associated with Mykoplasma infections. 

36. Mycoplasma culture- SP4 liquid and media need a manufacturer- maybe also some detail on the type of agar media?

The media was self-made as described previously. The text provides the reference how to produce the media: it states: “The samples were cultured using SP4 liquid and agar media as described previously [34].” We added “produced in house” for clarification.

37. Liquid and solid media were inoculated- which ones?

Both: This has been added.

38. In the case of colour change, or after five days, an additional …. (add in commas). Done

39. Mycoplasma needs capitals and italics. Done

40. Mycoplasma PCR- PBS needs a manufacturer

PBS does not need a manufacturer as we make this in-house- it is very clear what PBS is as it is a standard tool.

41. Not good to start a sentence with 100ul. 

Sentence was changed as proposed

42. Again, Mycoplasma needs capitals and italics. Done

43. First line of second paragraph doesn’t make sense- it is because of the reference so maybe reword it? 

Done

Results

44. Why were 11 chicks gender unknown? Too young to tell? Has been changed

45. You mention some haemorrhages in the pathomorphological section- could these have been caused during trapping? Yes

46. 2nd paragraph of pathomorphological section- line 5. Comma after ac3

Line 2 of paragraph 3- comma after ac3. Done

47. You mention nematodes seen in the pheasants- were these further identified? No, unfortunately not. 

48. Line on intestinal mucosa displayed in all animals- reword as unclear. Done

49. Coccidia needs capitalising and italicizing. Done

50. Were the protozoans not further identified either? No, unfortunately not.

51. Sarcocporidia- needs capitalising and italicizing. To our knowledge this is not the case. 

52. Virology section- comma after tested in first line. Done

53. Line 2- remove positive by. Done

54. Bacteriology section – you did 13 tracheal swabs from the 23 chicks- why was this number smaller and not one done from each?

The samples were only taken for Mykoplasma investigations. A total of 21 samples were taken. This has been clarified in Material and Methods. 

55. Third line- remove that. Done

56. You say via molecular biological methods- which ones- explain them please so we know

In the Material and Methods section we described:

All samples and single colony subcultures were screened via Mycoplasma-genus-specific PCR (target: 16S rRNA gene sequence) for DNA of Mycoplasma spp. as described by [37] and modified [36]. From all single colony subcultures, an additional PCR (target: 16S-23S rRNA sequence (Intergenetic Transcribed Spacer Region)) was performed [38]. Furthermore, all samples were examined via Mycoplasma gallisepticum-specific PCR [39]. The PCR products were sequenced by a commercial DNA sequencing service (LGC Genomics GmbH, Berlin, Germany). The sequences of the PCR products were aligned with the 16S rRNA gene and 16S-23S rRNA ISR sequences of Mycoplasma spp. in the NCBI database using BLAST (NCBI, USA) algorithm [40].

57. Comma after fifteen on last line of first paragraph. Done

58. First line of 2nd paragraph- remove were. Done

59. 2nd line of 2nd paragraph- Mycoplasma wants capitals and italics. Done

60. Parasitology- Line 3- spaces around the Ascaridia spp. Done

Discussion

61. Line 1- observed has a typo. Done

62. Line two- decline increased sounds a bit strange- consider rewording. Has been changed

63. Last two lines of paragraph one do not make sense and need rewording. Done

64. Paragraph 2- line 7- comma after alterations. Done

65. Line 7, remove as from start of sentence and put M gallisepticum in full. Done

66. Line 9. Full stop after [42] and start a new sentence. Done

67. Line 11- mycoplasma in italics. Done

68. Penultimate line of paragraph 2- last but not least sounds colloquial., maybe finally? Has been changed

69. Paragraph 3, not good to start it with a number. Has been changed

70. Paragraph 5- resulting in Dyspharynx nasuta – doesn’t make sense, consider rewording. Has been changed

71. Also last reference of paragraph 5 in capitals and incorrectly cited

And references in paragraph 6 are the same- maybe a glitch with your reference management software. Has been changed

72. Conclusions- line Line 7- hyphen seems inappropriate. Has been changed

73. End of first conclusion paragraph- maybe state that this hypothesis hasn’t been confirmed yet in pheasants. Has been changed

74. Table 4- I feel that this is a bit too big for a paper. Could this be supplementary? Yes, we agree that the table is large and we have now changed it to supplementary material (S1 Table).

---

## [Editor Report · Decision Letter 1]

19 May 2020

Health status of free-ranging ring-necked pheasant chicks (Phasianus colchicus) in North-Western Germany

PONE-D-19-33260R1

Dear Dr. Siebert

We are pleased to inform you that your manuscript has been judged scientifically suitable for publication and will be formally accepted for publication once it complies with all outstanding technical requirements.

With kind regards,

Simon Clegg, PhD

Academic Editor

PLOS ONE

Additional Editor Comments (optional):

Many thanks for resubmitting your manuscript to PLOS One

I have reviewed your manuscript, and as you have addressed all the points raised in the initial review, I have recommended the manuscript for publication

You should hear from the Editorial office

It was a pleasure working with you, and I wish you all the best for your future research

Hope you are keeping safe and well in these difficult times

thanks

Simon

---

## [Editor Report · Acceptance letter]

26 May 2020

PONE-D-19-33260R1 

Health status of free-ranging ring-necked pheasant chicks (*Phasianus colchicus*) in North-Western Germany 

Dear Dr. Siebert:

I am pleased to inform you that your manuscript has been deemed suitable for publication in PLOS ONE. Congratulations! Your manuscript is now with our production department. 

With kind regards,

on behalf of

Dr. Simon Clegg 

Academic Editor

PLOS ONE